# InSAR-Informed In-Situ Monitoring for Deep-Seated Landslides: Insights from El Forn (Andorra)

Rachael Lau[1], Carolina Seguí[1], Tyler Waterman[1], Nathaniel Chaney[1], and Manolis Veveakis[1]

[1]Department of Civil and Environmental Engineering, Duke University, Durham, USA.

**Abstract.** Monitoring deep-seated landslides via borehole instrumentation can be an expensive and labor-intensive task. This work focuses on assessing the fidelity of Interferometric Synthetic Aperture Radar (InSAR) as it relates to subsurface ground motion monitoring, as well as understanding uncertainty in modeling active landslide displacement for the case study of the in-situ monitored El Forn deep-seated landslide in Canillo, Andorra. We used the available Sentinel-1 data to create a velocity map from deformation time series from 2019-2021. We compared the performances of InSAR data from the recently launched European Ground Motion Service (EGMS) platform and the ASF On Demand InSAR processing tools in a time series comparison of displacement in the direction of landslide motion with in-situ borehole-based measurements from 2019-2021, suggesting that ground motion detected through InSAR can be used in tandem with field monitoring to provide optimal information with minimum in-situ deployment. While identification of active landslides may be possible via the use of the high-accuracy data processed through the EGMS platform, the intents and purposes of this work are in assessment of InSAR as a monitoring tool. Based on that, geospatial interpolation with statistical analysis was conducted to better understand the necessary number of *in-situ* observations needed to lower error on a remote-sensing recreation of ground motion over the entirety of a landslide, suggesting between 20-25 total observations provides the optimal normalized root mean squared error for an ordinarily-kriged model of the El Forn landslide surface.

## 1  Introduction

Deep-seated landslides represent one of the most devastating natural hazards on earth, many creeping at inappreciable velocities over several years before suddenly collapsing, usually with catastrophic velocities (Smalley, 1978; Voight, 1988). While there are a range of landslide sizes, several deep-seated landslides include sizable earth slides involving millions of cubic meters of soil moving as a rigid block on top of a deep (below the roots of the trees and the groundwater level) basal layer of heavily deformed minerals (Petley and Allison, 1997; Frattini and Crosta, 2013). Their collapse is usually very sudden, happening within minutes and without a clear warning, reaching high velocities, as high as the 20 m/s reported at the 1963 Vaiont landslide in Italy (Voight, 1988; Smalley, 1978; Veveakis et al., 2007). The catastrophic and fast collapse of this kind of landslides makes the evacuation of the area that could be affected a cumbersome task, thereby increasing risk of fatalities and infrastructure damages (Reid, 1994; usg; Huang et al., 2011; Guzzetti, 2000). Moreover, the complex physical nature of the landslides induces high uncertainty in the number of *in situ* observations required for a high-fidelity monitoring system.

That, in combination with the challenging and expensive methods of *in situ* monitoring, makes the development of reliable, data-driven, early warning systems (or tools/protocols to stop the acceleration of the landslide) an appealing proposition.

Before the use of satellites, initial approaches in predicting the catastrophic collapse of a landslide rely on physical access in the area with in-situ (extensiometer) or ex-situ (LiDAR, UAV) displacement data, whereby an assessment is made by using the inverse velocity method (Jaboyedoff et al.; Yaprak et al.; Saito, 1969; Carlà et al.; Zhou et al., 2020). Considerable work has since been done in developing remote sensing methods for landslide identification (Handwerger et al., 2019; Zhong et al., 2020; Mohan et al., 2020; Casagli et al., 2023; Chi et al., 2002; Zhao and Lu, 2018) as well as creating predictive models of deep-seated landslides based on identifying different mechanisms involved as triggering factors of the acceleration like rainfall (Reid, 1994), temperature (Mitchell et al., 1968; Veveakis et al., 2007) and chemical alterations (Hueckel and Pellegrini, 2002). Both developments are now at a stage were they can be used in conjunction with high fidelity field data (piezometers, extensiomenters and thermometers) to obtain forecasting and mitigation protocols (Seguí and Veveakis, 2021, 2022). However, the installation of such in-situ instrumentation is a costly operation, requiring the transportation of heavy equipment often in remote areas and the installation of sensors in deep boreholes, that cannot be deployed readily across the world.

To overcome these constraints, the use of remote sensing has become a more available tool for landslide monitoring over the last several decades. Several techniques for mapping and assessing slope movements have been developed, thus allowing for more reliable and fast investigation (Cigna et al., 2013; Fiorucci et al., 2011; Guzzetti et al., 2009; Michoud et al., 2012). Among the remote sensing options, the use of Synthetic Aperture Radar (SAR) sensors has gained significant popularity for measuring surface deformations and constructing their time series, since this approach requires no access to the site to install borehole instrumentation or handle UAV and LiDAR devices. Remote monitoring approaches for deep-seated landslide are limited by their often inability to provide information for the body of the landslide when the moving mass is deep-seated in steep valleys or densely vegetated mountain ranges, as well as their nature as surface-only measurements. This work builds on the existing literature concerning the assessment of how reliable remote surface measurement tools could be for deep-seated landslides (Bayer et al., 2017; Fobert et al., 2021; Bellotti et al., 2014; Casagli et al., 2023; S and Kanungo, 2004; Lissak et al., 2020; Scaioni et al., 2014; Wang et al., 2019), providing a case study from the El Forn landslide in terms of the data quality needed to identify and monitor a landslide and extending this body of literature by using InSAR data to decide the minimum number of in-situ observations needed for that.

## 2 Material and Methods

### 2.1 Description of the El Forn Landslide and In-Situ Data

The El Forn landslide is a large deep-seated landslide located southeast of the town of Canillo, Andorra, nestled in the Pyrenees (see Figure 1) that is triggered by snow melt and season rainfall that collect into an aquifer located below the sliding surface. This landslide has a sliding mass of approximately 300 $Mm^3$ that creeps at an average rate of 1.2 cm/year (Seguí and Veveakis, 2021). Within the main sliding mass of the landslide, there is a faster-moving lobe (Cal Ponet-Cal Borronet lobe) that slides at a maximum velocity of 2-4 cm/year (EuroConsult; Seguí and Veveakis, 2021; Zhao and Lu, 2018). At present, this lobe is

equipped with 12 boreholes dispersed between the top and bottom of the landslide collecting continuous in-situ data. However, S10 is the only *continuously-monitored* borehole on the landslide, with measurements every 20 minutes. Other boreholes are monitored via analog non-continuous measurements (irregularly, approximately once per month), which is why we chose to work exclusively with S10. It is important to note that, while there is available analog data over the landslide, only one point is a viable option for comparison with InSAR, given the no-snow period of 2019. In order to not reduce the fidelity of the continuous time series from S10, the authorship chooses to not include this data in the body of this text. The authorship has made this data available upon request.

The sliding surface is located at 29m depth, and the landslide is moving as a rigid block (Seguí and Veveakis, 2021) on top of it, creeping into the town of Canillo, as shown in Figure 1, with periods of greatest acceleration during the no-snow periods (May-August) each year. The shearband is comprised of 80% Silurian shales rich in phyllosilicates (muscovite, paragonite, and chlorite), and 20% quartz (Seguí et al., 2020). A more detailed explanation of the geological makeup of the shear band can be found in Seguí et al. (2020). The terrain of the landslide can be seen in Figure 1.

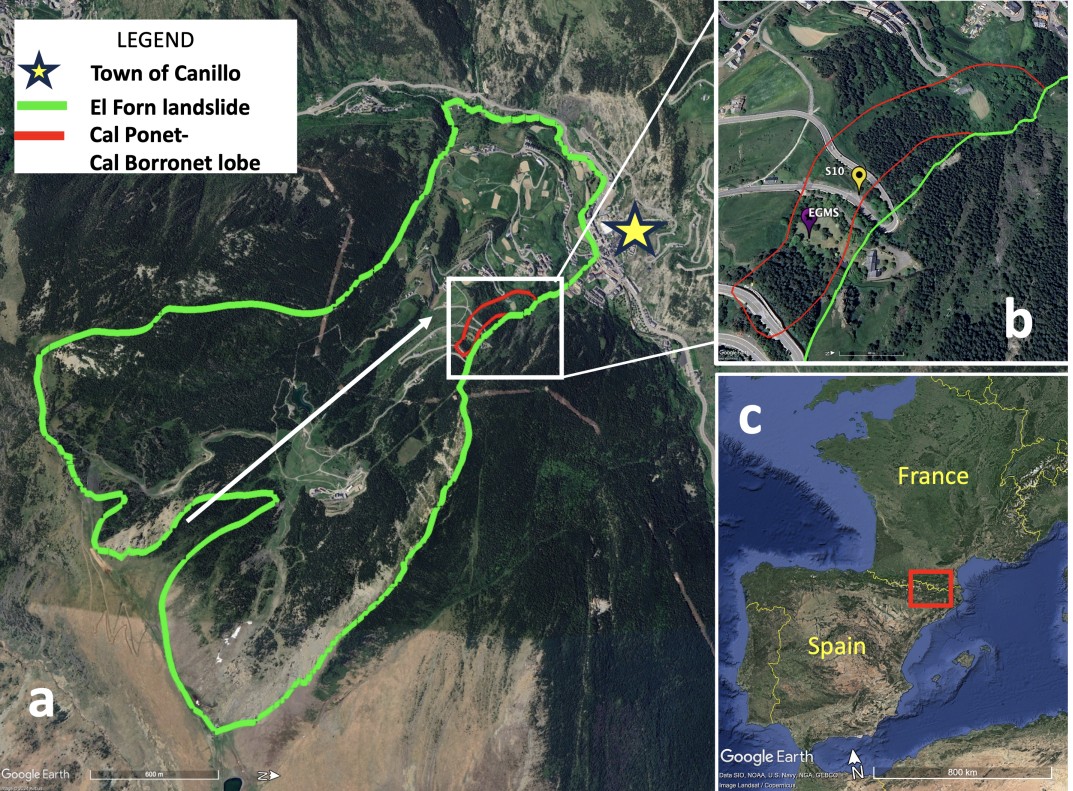

**Figure 1.** (a) Overview of El Forn landslide with Cal Ponet-Cal Borronet lobe, noted with EGMS observation (see section 2.2.2) and S10 borehole location. White arrow shows direction of landslide into the town of Canillo, marked with a star; (b) Cal Ponet-Cal Borronet lobe with S10 borehole; (c) Localization of the landslide in Andorra. Images ©2024 Airbus.

The primary instrumentation and data considered in this study are housed within borehole S10, noted as the yellow marker in Figure 1. Data from S10 is sampled continuously every 20 minutes via instrumentation including an extensometer, three piezometers, and a thermometer within the shear band, which measure horizontal displacement, water pressure, and temperature changes in the material, respectively. The data considered in this study for El Forn are displacement gathered from the extensometer. Detailed context for the location of S10 and the depth profile of the landslide with S10 borehole readings can be found in Seguí and Veveakis (2021).

## 2.2 Remote Data Collection and Processing

One of the key objectives of this work is to compare InSAR to subsurface ground measurements. This is achieved through interferograms obtained by Sentinel-1 A/B over a period of 6 months in 2019 with a 6 day acquisition interval. It is important to note that the landslide was arrested at the end of 2019, so 2019 remains the year of focus for the intents and purposes of this work. Additionally, a 6 month InSAR time interval is chosen in order to avoid snow cover seasons on El Forn since backscatter from snowcover makes use of InSAR particularly difficult due to low coherence (Kumar and Venkataraman, 2011; PBC, 2022). Interferograms are then processed to obtain displacement time series over the landslide's surface using two different approaches:

(1) a high-precision (fine spatial resolution), low-accuracy (noisy) approach whereby Sentinel-1 data is retrieved and preprocessed with a low coherence threshold to obtain high spatial resolution (40x40 meter grid) displacement data so that geospatial analysis can be conducted to determine the minimum number and location of observations required for landslide monitoring and reconstruction with quantified uncertainty. This approach was deployed using the Alaska Satellite Facility's (ASF) Vertex Platform's On Demand InSAR processing tools and will be hereinafter referred to as ASF; and

(2) a low-precision (sparse spatial resolution), high-accuracy (de-noised) approach whereby Sentinel-1 data are filtered to reduce the noise so that landslide identification can be achieved from high-accuracy data on a 100x100 meter grid. This approach was performed via immediate download through the newly-launched European Ground Motion Service (EGMS) Platform by Copernicus (EGM) and will be hereinafter referred to as EGMS.

Note that the SAR imagery for both the data retrieved via the ASF On Demand InSAR processing tools and the Copernicus EGMS portal was taken from Sentinel-1 A/B satellites on a descending track with a 270-degree angle of incidence from the vertical. Using the slope of the ground at S10, the data for the EGMS displacement and ASF-MintPy readings were translated into the displacement along the direction of the landslide movement so it could be compared to S10's strain gauge readings.

While displacement data from EGMS is readily-available, data retrieval from ASF requires a more hands-on approach, going through a short baseline subset pre-processing step via ASF On Demand InSAR processing tools, followed by an interferogram time series inversion via the Miami InSAR time-series software for Python (MintPy) that allows us to generate mean deformation velocity maps and deformation time series (Berardino et al., 2002; Handwerger et al., 2019; Yunjun et al., 2019). Subsequent displacement data from this time series inversion, alongside displacement data pulled from the EGMS platform are compared with in-situ displacement data from S10 to understand correlation between InSAR and in-situ data. The other key objective of this work is to understand how InSAR can be used for general uncertainty quantification for planning future bore-

105 hole placement, should the first objective prove InSAR can be correlated with sub-surface measurements. This will be done via iterative ordinary kriging, with the normalized root-mean-squared-error (RMSE) being used as the statistical parameter of interest for confidence. The next paragraphs outline the technical details of data retrieval and processing for each of these approaches.

### 2.2.1  ASF InSAR Data Retrieval and Time Series Inversion

Open-access descending-track SAR acquisitions from Sentinel-1 C-band (approximately 5.66 cm radar wavelength) were pulled from the Alaska Satellite Facility's (ASF) Vertex Portal and processed automatically through this portal via the Advanced Rapid Imaging Analysis (ARIA) for Natural Hazards Project (Bekaert et al., 2019).

InSAR data retrieved for the purposes of this work were retrieved by selecting Single Look Complex (SLC) scenes with a beam mode of Interferometric Wide (IW) and cover the El Forn landslide. Using the Alaska Satellite Facility's On Demand
tool, scenes were selected and pre-processed using the Short Baseline Subset (SBAS) tool, making it easier to order the best interferograms for SBAS. However, MintPy's default time-series tool was ultimately used. From there, all 619 interferograms covering El Forn were downloaded via Python script from the ASF Vertex Platform. Interferograms with visible discontinuities were manually identified once downloaded and removed from the stack for time-series analysis. Please see supplementary material S1 for more information on InSAR pre-processing and subsequent workflow. From there, the ASF Hybrid Pluggable
Processing Pipelines (HyP3) service allowed for each interferogram to be clipped to the same size of overlap to standardize each interferogram. Using MintPy (see Yunjun et al. (2019) for more information on the time series inversion process), clipped interferograms were then inverted to create a deformation time series using a weighted least squares inversion with a coherence threshold value of 0.4. This approach creates velocity and deformation maps on a 40x40 meter grid, as shown in Figure 2(a). It is to be noted that the low coherence threshold used provides high spatial resolution maps that can be used for geospatial analysis,
however this increased resolution is accompanied by increased noise, which makes landslide identification cumbersome, as seen in Figure 2(a) and discussed in the results section. For this reason, a second approach is pursued in parallel, focusing on the accuracy of the data as detailed below.

### 2.2.2  EGMS InSAR Data Retrieval and Time Series Inversion

A second set of InSAR data was taken from the European Union's Copernicus project via the European Ground Motion
Service (EGMS) portal, available for immediate retrieval as vertical and East-West displacement series per point. This platform provides already-processed displacement data over parts of El Forn at a grid of around 100x100 meter resolution as seen in Figure 2(b), with the location of data points apparent as dots over the topography of the El Forn landslide.

As already mentioned, there are key differences between the data retrieved via the ASF On Demand InSAR processing tools and the data retrieved from the EGMS portal for the intents and purposes of this work – the key trade-off being between
135 precision and accuracy. More specifically, the ASF On Demand data inverted via MintPy used a minimum threshold coherence value of 0.4, whereas EGMS Ortho Data only visualized individual measurement points greater or equal to 0.8. It is to be noted that the EGMS results could have also been obtained by applying a higher threshold value to the ASF approach, making the

choice between the tool used scientifically immaterial. The reasons for utilizing both in parallel are the ability to showcase and cross-validate the two approaches.

## 2.3 Spatial Interpolation and Ordinary Kriging

Ordinary kriging was conducted by first creating a grid of $x$- and $y$- coordinates and corresponding velocity values at these points. Distances between the random observations and each individual grid point were calculated, such that:

$$d_1 = \sqrt{(x_g - x_{obs}^T)^2 + (y_g - y_{obs}^T)^2}, \tag{1}$$

where $x_g$ and $y_g$ are the grid coordinates, and $x_{obs}$ and $y_{obs}$ are the random observation coordinates. The covariance matrix were determined using the range $\tau$ and variance $\sigma^2$ from the semivariogram, such that:

$$C = \sigma^2(e^{(-d_1/\tau)^T}) \tag{2}$$

The euclidean distances between the random observations and each other were calculated, as well as the corresponding covariance matrix $\Sigma$, such that:

$$d_2 = \sqrt{(x_{obs} - x_{obs}^T)^2 + (y_{obs} - y_{obs}^T)^2} \tag{3}$$

$$\Sigma = \sigma^2(e^{(-d_2/\tau)^T}) \tag{4}$$

The covariance matrices were appended into two matrices that would be used Lagrange Multipliers, into matrices $\Sigma'$ and $C'$, respectively. The weights were calculated by solving the linear equations created by the $\Sigma$ and $C$ matrices. From there, we calculate predictions $Z^*$ by taking the values velocity values at the random observations, $z_t$, multiplying them by the corresponding weights $W$, such that:

$$Z^* = \Sigma(W * z_t) \tag{5}$$

The mean squared error is then solved, such that:

$$MSE = \sigma^2 - \Sigma(W * C') - W \tag{6}$$

As a result, fidelity was assessed via root mean-squared error ($RMSE = \sqrt{MSE}$) of a kriged landslide surface done via random sampling done without replacement per iteration.

# 3 Results

## 3.1 Landslide Identification

As previously noted, ASF and EGMS data showcase a key trade-off between precision and accuracy for the purposes of landslide monitoring. Figure 2(a) demonstrates that with a lower coherence value, more data is available, albeit noisy. In a separate vein, the EGMS data (pictured in Figure 2(b) demonstrates the utility of an increased coherence value in reducing
 noise and producing high-accuracy data usable for landslide detection.

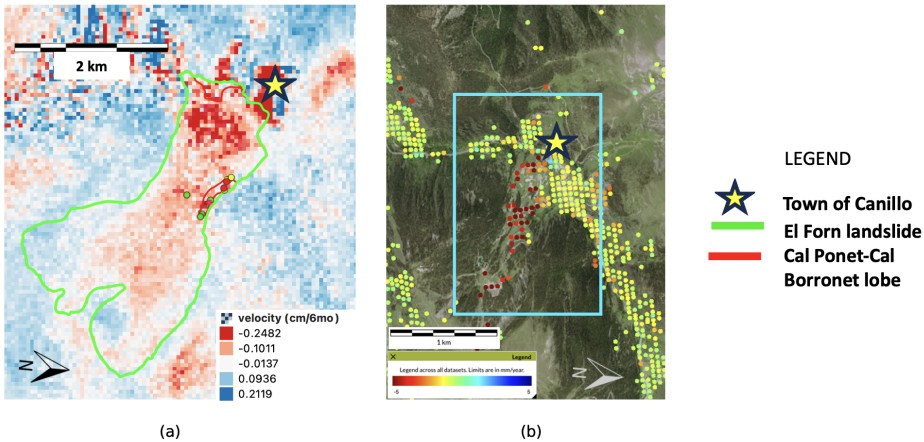

**Figure 2.** Depiction of El Forn landslide using InSAR. (a) Overview of El Forn landslide velocity using data retrieved and inverted using ASF against the field observation of the landslide boundary (green line). Colored dots indicative of total borehole locations, delineated by color due to separate monitoring agencies in partnership with the Government of Andorra.(b) InSAR detection of El Forn landslide by Copernicus EGMS platform (highlighted in blue), retrieved 28 September 2023. Indicative of possible use of EGMS as tool for active landslide detection.

## 3.2 Correlating InSAR with In-Situ Data for seasonal ground motion

Upon data retrieval, both ASF and EGMS InSAR displacements were compared along the direction of sliding with *in situ* strain gauge data from borehole S10 in order to understand the fidelity of InSAR in monitoring sub-surface ground motion. This direct comparison of InSAR readings from ASF and EGMS over the S10 borehole can be seen in Figure 3. Retrieval of
 InSAR displacement data from EGMS required manual comparison of a couple of neighboring points with S10's *in situ* data in order to find points with a strong enough signal to use to compare since there was no individual measurement point at the location of S10 after the increased threshold was applied (see Figure 1).

Indeed, the increased sparsity from EGMS resulted in a lack of precision of the individual measurement points to compare with *in situ* measurements, as seen in Figure 3(c), as compared to data retrieved via ASF's On Demand tools (as seen in Figure

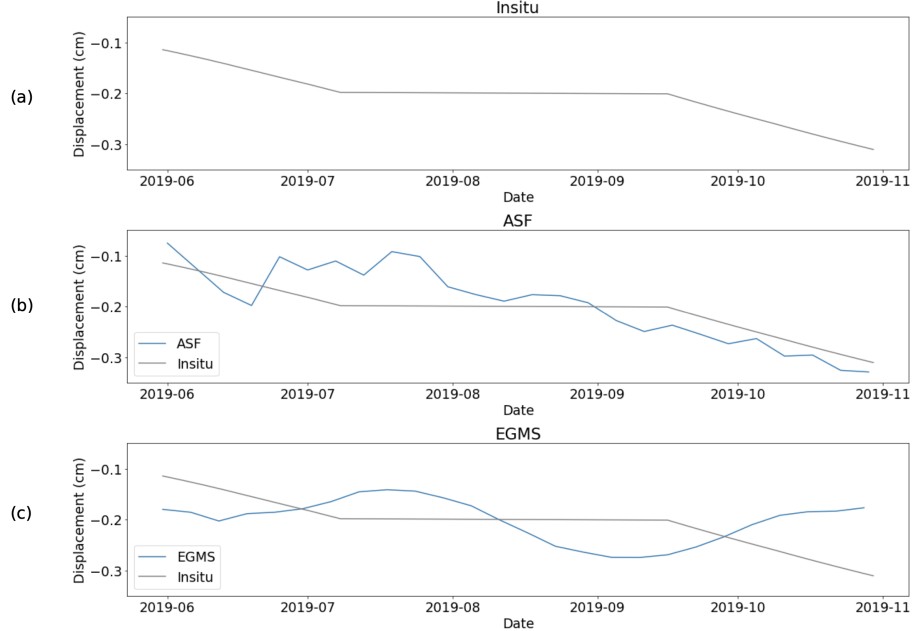

**Figure 3.** Comparison of *in situ* displacement data with displacement data retrieved via EGMS and ASF On Demand Processing tools. (a) *in situ* displacement readings from S10 borehole. (b) 7-day cumulative moving average of InSAR displacement readings over S10 with data retrieved via ASF On Demand Processing tools. (c) 7-day cumulative moving average of InSAR displacement readings with data retrieved from EGMS.

3(b). Since data for the exact location of S10 borehole on the landslide was not immediately available on the EGMS platform, two coordinates neighboring the WGS-84 coordinate of S10 were pulled and compared to the S10 data and InSAR data. Heterogeneity within the landslide prevented selecting just one point as close to the S10 point as possible, without properly examining other neighbors. Figure 1 details which point was examined, with "EGMS" being the point in the EGMS database that was ultimately used because of its closest alignment with S10's raw displacement data. Figure 3 directly compares data retrieved with ASF On Demand Processing tools (and inverted via MintPy) and EGMS with *in situ* displacement measurements. We observe that while InSAR displacement measurements pulled from EGMS are helpful in detection, the higher accuracy creates a lack of precision necessary for *in situ* comparison.

In order to justify this claim and quantify the performance of the two approaches in time, a measure of linear independence (correlation) was conducted with data from EGMS and ASF with *in situ* measurements, respectively. The equation used for the calculation of the correlation coefficient $\rho(A, B)$ of two datasets $A$ (in this case EGMS or ASF) and $B$ (in this case the *in situ* data) is:

$$\rho(A,B) = \frac{1}{N-1}\sum_{i=1}^{N}\left(\frac{A_i - \mu_A}{\sigma_A}\right)\left(\frac{B_i - \mu_B}{\sigma_B}\right), \tag{7}$$

where $\mu$ and $\sigma$ are the mean and standard deviation of the data sets, respectively. The results of the performance of ASF and EGMS against the *in situ* data are detailed in Table 1, where we see that the ASF dataset ($\rho(ASF, in\ situ) = 0.6957$) has a

considerably better performance against *in situ* data than EGMS ($\rho(EGMS, in\ situ) = 0.0761$). This is presumably due to the spatial heterogeneity of the landslide's displacement around the point of measurement, which is not exactly on S10 (see Figure 1), and is not reflective of the overall quality of the EGMS data.

**Table 1.** Comparison of Correlation Coefficients (Eq.7) of displacement data retrieved from ASF and EGMS with *in situ* measurements.

|  | in situ | ASF | EGMS |
|---|---|---|---|
| *in situ* | 1.000 | 0.6957 | 0.0761 |
| ASF | 0.6957 | 1.000 | - |
| EGMS | 0.0761 | - | 1.000 |

### 3.3   Ordinary kriging: determining necessary number of remote observations

Having shown the correlation between ASF and *in situ* in the previous section, we move forward with the densely-populated

ASF mapping to carry out ordinary kriging. In order to best understand how many observations (i.e., boreholes) impacts the ability to remotely model ground motion of a deep-seated landslide, 200 iterations of randomly-selected samples (with sizes ranging from 5-100 points) along the main landslide surface were selected and had ordinary kriging performed on them to assess the RMSE in predicting ground motion over the surface of the landslide, with summaries of RMSE for each number of iterations visible in the box and whisker plot in Figure 4. Similarly, single-iteration ordinary kriging was conducted over the

sliding mass to assess how various sample sizes (ranging from 10-2000) were in *recreating* velocities of the sliding mass, and where possible areas of interest for further investigation were.

More specifically, 200 iterations were conducted per number of random observations of the average velocity in 2019 (pulled from a uniform distribution, as seen in Figure 4b), results of which can be seen in Figures 4c. Figure 4c, the box for the normalized root mean squared error (RMSE), indicates a marked drop in the interquartile range (IQR) between 20 and 25

observations. Also important to notice is that the range of outliers is significantly lower starting from n = 25 observations and forward. Note that the dots outside of the whiskers are outliers, meaning they lie outside of the whiskers defined by $Q1 - 1.5 * IQR$ and $Q3 + 1.5 * IQR$.

Figure 5 reflects how *n* samples recreates the avreage landslide velocity movement over the no-snow periods in 2019 in the line of sight (LOS) and the fidelity (RMSE) of doing so. For example, n = 30 random samples recreates certain parts of the

landslide better than others for one iteration. Figure 5 shows the evolution of how increased random samples that go through ordinary kriging process then recreate certain parts of the landslide surface faster or better than others. In the case of n =

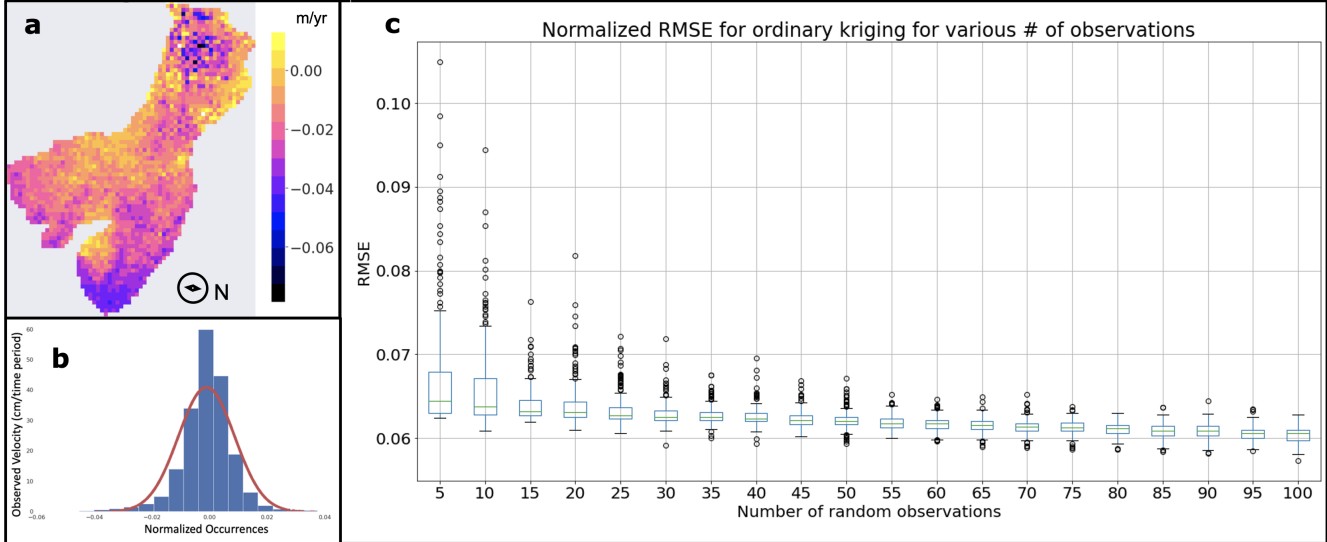

**Figure 4.** (a) InSAR velocity map of the 2019 snow-free period in Andorra via ASF-processed InSAR, (b) uniform distribution probability density function (red line) and occurrence histogram (blue) of velocities pulled from sub-figure (a), (c) boxplot of normalized RMSE of 200 iterations for various number of random observations of velocities from sub-figure (a) pulled from uniform distribution in sub-figure (b).

30, the top and bottom of the landslide are better developed than the middle of the landslide, which indicates where further investigation may be necessary. More specifically, the center of the landslide is the least developed throughout the ordinary kriging iterations – for modeling purposes then, further investigation would be required on this part of the sliding mass (either further instrumentation or a more narrow scope of InSAR).

## 4 Discussion

The application of Interferometric Synthetic Aperture Radar (InSAR) for deep-seated landslide monitoring represents a significant advancement in geohazard assessment and management. The use of InSAR and its comparison to traditional *in situ* approaches for sub-surface ground motion serve as important next steps in assessing the viability of the remote sensing tool for large-scale deep-seated landslide monitoring. However, there are known limitations in this approach, including limited sub-surface *in situ* borehole readings to directly compare with InSAR. Additionally, as seen in Figure 3, there is a trade-off between accuracy and precision when it comes to the use of InSAR for landslide detection versus monitoring. However, for considering the use of InSAR as a *monitoring* tool, this approach is limited in the assumption that the deep-seated landslide moves as a rigid block, as opposed to other deep-seated landslides that may move sequentially, and not as uniformly. In the latter case, other sub-surface *in situ* measurements may be necessary to verify the movement of the landslide. There are, conversely, several advantages to the use of InSAR as a sub-surface monitoring tool, including its possibility to be linked with existing deep-seated landslide models (Veveakis et al., 2007; Seguí and Veveakis, 2022; Lau and Veveakis, 2024). The ability to correlate InSAR

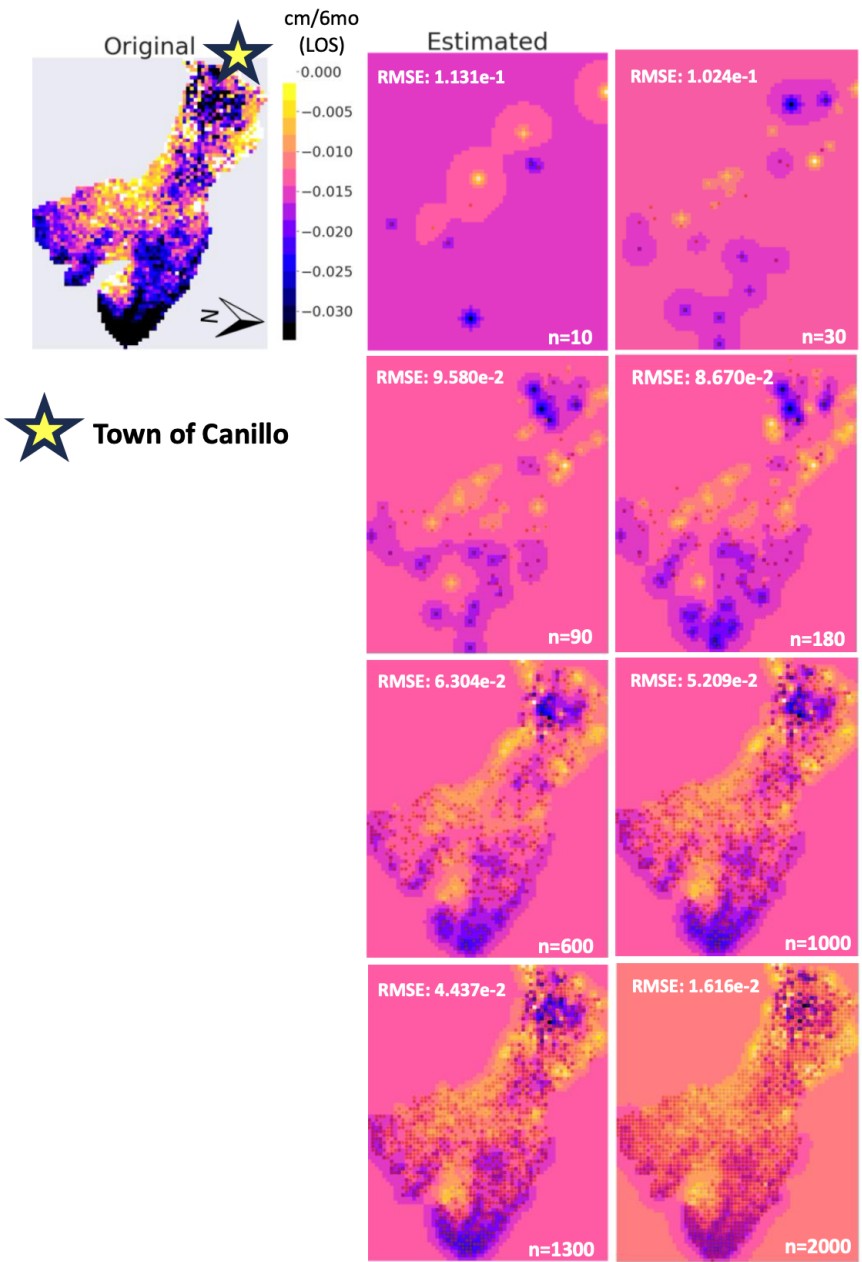

**Figure 5.** Ordinary kriging results of various random samples (n = 10 - 2000) via one iteration (as opposed to the 200 iterations of average velocity values (during the no-snow months of 2019 in the direction of the line of sight (LOS)) in Figure 3(c)), reflecting gaps in predictive capabilities on the surface of the landslide for further investigation. Normalized RMSE for each ordinary kriging process indicating error for each sample size. The town of Canillo is marked with a star and cardinal directions have been added for context.

displacement readings with sub-surface ground motion, as addressed in this work, lends itself to be applied to well-developed models in which displacement from InSAR and *in situ* borehole readings can tune stability models for deep-seated landslides that use temperature in the landslide (frictional heating) as the primary driver for tertiary creep and catastrophic collapse of these large mass movements. With that in mind, there is an opportunity to apply InSAR as a *forecasting* tool. This opportunity, explored more in-depth in Lau and Veveakis (2024), has the potential to lead to a majority- or in some cases completely-remote approach to deep-seated landslide forecasting, utilizing InSAR and existing borehole data to develop and tune existing models deep-seated landslide stability (Veveakis et al., 2007) .

It is important to note, however, that the authorship acknowledges that the best landslide monitoring options often derive data from a variety of sources – remote sensing, *in situ* instrumentation, and narrative accounts. However, unlike ground-based methods such as borehole instrumentation, which can be labor-intensive and expensive, InSAR provides comprehensive spatial coverage, with the capability to be applied to to monitor remote or inaccessible regions. This is particularly beneficial to rural mountain communities who constantly face some degree of exposure to deep-seated landslides, and who often reside in rugged terrains where installing and maintaining ground-based instruments can be challenging, costly, or financially inaccessible.

Of course, the results completed in this work could always be enhanced by the addition of more *in situ* monitoring options for the El Forn landslide to offer more direct comparison between InSAR and borehole displacement readings. Similarly, this work could be enhanced by more landslide case studies, perhaps with slopes facing other directions in order to better understand the sensitivity of *in situ* and InSAR data to the way the InSAR data was taken. Both of these possibilities for improvement are considered by the authorship for future works.

## 5  Conclusions

In this paper, the use of InSAR for landslide monitoring was assessed for two key objectives: (1) correlation with *in situ* data to test the accuracy of InSAR in monitoring seasonal and off-seasonal sub-surface movement, and (2) spacial interpolation across the landslide surface with various number of InSAR points to help us understand use of InSAR for establishing areas on a scarp that need monitoring (i.e. further instrumentation), as well as understanding how many remote observations would allow us to minimize error in recreating the scarp without using the full data set. Correlation of the InSAR data with extensometer data in S10 borehole on the El Forn scarp indicates that InSAR can be used to understand seasonal sub-surface ground motion.

The spatial interpolation, as well as the susbequent error assessment, conducted on the El Forn landslide using solely InSAR data helped determine the necessary number of observations to adequately monitor the general movement of the landslide. Based off of 200 iterations of random samples going through an ordinary kriging process on the landslide, the outliers of the normalized root mean squared error dropped significantly between 20 and 25 remote observations, as indicated in Figure 4. Based off of Figure 5, the most uncertainty, coupled with the most movement, through even an increased number of random samples is in the middle of the landslide, can be seen in the middle of the top left lobe, in the northeast corner of the landslide. In future studies, we could look to perform regression kriging with 20-25 remote observations, focused solely in this region to understand how uncertainty propagates for this part of the landslide on a finer time scale. Overall, InSAR has many purposes

when considering the monitoring of deep-seated landslides, with several options to build on our existing knowledge for studies to come.

*Data availability.* Data sets are open-access and retrievable from the Alaska Satellite Facility Vertex Platform (https://asf.alaska.edu/) and the European Ground Motion Service Copernicus Platform (https://egms.land.copernicus.eu/).

*Sample availability.* *In situ* sample data for S10 is available at insitudata. Additional data is available upon request of the corresponding author.

## Appendix A: Alaska Satellite Facility (ASF) InSAR Workflow

The Alaska Satellite Facility's Vertex Platform user-friendly interface allows for ease of specifications on selecting an InSAR pair. For this work, an interferometric-Wide Single Look Complex (IW SLC) pair was selected – SLC meaning that the SAR

data has been compiled to an image but hasn't been multi-looked yet. Once the reference and dates of interest are selected, ASF begins multi-looking through various pairs of images. Note that for continuity purposes, the older SLC image is always used as the reference image.

In order to prepare a digital elevation model (DEM) file for subsequent geocoding and corrections, a topographic phase is subtracted from the interferogram by replicating an existing DEM to account for the actual topographic phase. In this

case, Hyp3 takes the DEM from the publicly-available 2021 Release of Copernicus GLO-30 DEM library. Removing this topographic phase from the interferogram, the deformation signal is all that remains (Hogenson et al., 2024).

Left with a stack of wrapped interferograms, phase unwrapping uses a Minimum Cost Flow (MCF) triangulation method to assign multiples of $2\pi$ to each pixel, which restricts the number of $2\pi$ jumps in phases to regions where they may occur. Note that thermal noise and interferometric decorrelation can result in $2\pi$ phase discontinuities, which are known as "residues" –

280 these can be reduced via filtering. Filtering reduces phase noise and increases the accuracy of interferometric phase by reducing the number of interferogram residues(Hogenson et al., 2024).

After filter, a validity mask directs the unwrapping process by applying thresholds for coherence and amplitude (backscatter intensity) values for each image pair. For this work, this amplitude threshold is kept to 0.0, so coherence thresholds drive the masks. Coherence is estimated from a normalized interferogram, with a range from 0.0 to 1.0, with 1.0 being perfectly

coherent. Once coherent thresholds are applied, unwrapping will proceed relative to a fixed pixel point – one that should have a fixed pixel point. For this work, this point was selected as a rooftop in the town of Canillo at the foot of the landslide. This reference point is assigned an unwrapped phase value of 0 at this point, and every other pixel around it is then assigned a multiple of $2\pi$ with respect to that point.

Lastly, these pixels are reprojected from SAR slant range space into a map-projected ground range-space and exported from the GAMMA internal format to GeoTIFF format. These unwrapped interferograms are ready to be go through a time series inversion(Hogenson et al., 2024).

*Author contributions.* R. Lau conceived, designed, and carried out the analysis, as well as wrote the entirety of the manuscript. Under the Graduate Student Training Enhnacement Grant at Duke University, R. Lau worked under A. Handwerger where he provided critical resource guidance on the use of InSAR, ISCE+, and MintPy for this work. C. Segui provided helpful information regarding geophysical modeling and geology of the El Forn landslide. T. Waterman provided a helpful resource in getting set up and trouble shooting working on the computing cluster. N. Chaney provided space on his lab's computing cluster, as well as helpful guidance on spatial data analysis. M. Veveakis provided supervision and writing guidance throughout the entirety of the research process.

*Competing interests.* The author(s) declare(s) that there is no conflict of interest regarding the publication of this article.

*Acknowledgements.* We acknowledge the collaborative efforts of our co-authors for their expertise and contributions to this work. Financial support from the National Science Foundation and the Fulbright Program was instrumental in conducting our investigations. We also appreciate the cooperation of the Government of Andorra in providing crucial in-situ data measurements. These combined efforts significantly enhanced the quality and scope of our study.

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
