# Peer review of "InSAR-Informed In-Situ Monitoring for Deep-Seated Landslides: Insights from El Forn (Andorra)"

_EGUsphere, 2024_

## Author Comment (AC1)

**RC 1 Feedback**

1. I recommend including a m**ore detailed geomorphological, geological and stratigraphical description of the area interested by the El Forn landslide**. For a complete characterization of a landslide through InSAR data, it is essential to investigate the geologic context in more detail to understand the conditions under which it is developing.

The reason we had a brief description of the site is because characteristics of the landslide have been published extensively. However, we understand the importance of including a more detailed description of the landslide for thepurposes of serving as a stand-alone work. We are happy to include this in a revised version.

2. **I recommend extending this InSAR analysis to the entire landslide body and not just the Cal Ponet-Cal Borronet lobe sector.** Because chapter 2.1 describes the presence of 12 scattered boreholes in the landslide body for monitoring that should be exploited as a real opportunity for comparison with the InSAR data. The greatest strength of satellite interferometry is the ability to monitor large areas, here authors have focused only on a very small sector of a very large landslide, missing the most important information provided by the InSAR data.

We have an InSAR analysis of the entire landslide, as seen in Figures 4 and 5 of the initial manuscript. The reason we focus on the lobe is because of the presence of borehole S10, which is the only continuously-monitored borehole on the landslide, with measurements every 20 minutes. Other boreholes are monitored via analog non-continuous measurements (irregularly, approximately once per month), which is why chose to work exclusively with S10. That being said, we are happy to include a comparison of InSAR with said other boreholes outside the lobe in a revised version.

3. **I recommend expanding the monitoring period of InSAR data, as the abstract specifies that Sentinel-1 data processed for 2019-2021 has been exploited, while Chapter 2.2 explains that interferograms from only a narrow time period between June and November 2019 were used.** Again, this choice comes at the expense of one of the major strengths of the InSAR data, namely the possibility of providing long time series. Instead, focusing the analysis only to a 6-month time period and on a narrow area of the landslide appears as a serious limitation in the study. When analysing the behaviour of a landslide, it is a good practice to expand the analysis of the time series as much as possible in order to know as much information as possible.

We have a time series detailing a yearly comparison of *in situ* and InSAR displacements on S10 and we are happy to include them in a revision version.

4. I strongly recommend adding a chapter discussing the results before the conclusions. A chapter of discussion is essential for the explanation of the results and to understand applicability, advantages, and limitations of the proposed approach.

Our original thought in the writing of this work was to have a short communication, but we agree the reviewer that the manuscript would benefit from a more detailed discussion fo the results. We will include this discussion in a revised version.

About the figures I suggest:

- Figure 1: a geographical localization is missing. North arrow between the large-scale image of the landslide and the focus are not in the same direction. I recommend pointing the north arrow upward. In addition, for the purpose of characterizing the study area please add the location of all 12 boreholes.
- Figure 2: the deformation map on the left has neither north arrow nor scale, also in the legend there is no explanation of what the coloured circles are. Moreover, the figure on the right is just a screen captured by EGMS: I recommend downloading the data and reshaping the figure (you can use the EGMS-stream application to download the data).
- Figure 4: it is not clear why the representation of the landslide is now rotated 90°. I recommend defining a direction for the landslide representation and using it for all the figures in the paper.
- Figure 5: the colour scale of the ordinary kriging results of various random samples are always different, this makes it so that an immediate visual comparison between the figures is not possible.

We will make suggested edits to the figures mentioned above.

---

## Author Comment (AC2)

**RC 2 Feedback**

The paper presents an interesting work on the use of InSAR data for monitoring deep landslides and defining the minimum number of observations useful for ensuring a valid model of landslide evolution. Once data have been correlated with field measurements, the Authors focused on how they can be used to quantify the overall uncertainty in planning borehole placement and implementing in-situ monitoring.

The case study analyzed, located in the Andorran Pyrenees, is significant both in terms of geomorphological evolution and natural hazard, thus well suited for the purpose of the work. In the introduction section, the work is well presented and contextualized in the available literature; the purpose and scientific validity of the work are adequately pointed out. The methods and methodological flow are clear and well-explained. In my opinion, results are valid but should be better discussed to highlight the strengths of the work. In detail, the following points are suggested to be developed for publication.

1. **Although it is not the main focus of the paper, I believe that additional morphological and geological data of the study area should be provided.** If available, the Authors should state in more detail the geological framework and the material involved in the landslide process, the geometry of the body, and the type of activity (e.g., periods of greatest acceleration). Additionally, if possible, it would be interesting to see the temporal evolution of some of the monitored data (e.g., displacements or piezometric level) and perhaps relate them to external factors such as rainfall. I think that providing a more extensive characterization of the landslide would allow an easier understanding of the data presented in the subsequent paragraphs, as well as the choices made; also, it will improve the quality of the paper. If the Authors do not consider it necessary, they may still better mention previous works carried out on the landslide body exploring these issues.

   The reason we had a brief description of the site is because characteristics of the landslide have been published extensively. However, we understand the importance of including a more detailed description of the landslide for thepurposes of serving as a stand-alone work. We are happy to include this in a revised version.

2. In Section 3.2, I would suggest a **better discussion of the choice of using a 6-month time interval to compare in situ observations and InSAR data.** What are the reasons for this choice that do not allow for the analysis of annual trends? It could be the presence of snow cover or lack of monitoring data, either way, it needs to be specified and discussed in the text.

   We chose to use a 6-month time interval to compare in situ observations and InSAR data because of snow cover, and we will be sure to note that more explicitly in a revised version.

3. Reading the paper, **I found the lack of a general discussion highlighting the significance of the work, especially in terms of the use and applicability of the method. I think a "discussion section" could help to better emphasize the validity** of the results obtained as well as the achievement of the goals. In my opinion, some scientific issues and questions that could be addressed in this section would be: What are the main advantages and limitations over other landslide monitoring methods? Can this method be applied in remote areas where there are no active monitoring systems? How can the method be improved and applied to implement an in situ monitoring network?

The reason we had a brief description of the site is because characteristics of the landslide have been published extensively. However, we understand the importance of including a more detailed description of the landslide for thepurposes of serving as a stand-alone work. We are happy to include this in a revised version.

5. Figures should be better developed to help reading and understanding of the work. I would like to suggest these modifications to the Authors:
    - Figure 1: The orientation of the satellite images is not very clear; it would be better to point the north arrow upward to maintain the same visual angle in all the sketches presented. I find placing the EGMS point in this figure to be not very intuitive; I would move it to other subsequent figures and insert here the location of the 12 boreholes mentioned in the text. Also, a small legend with the meaning of the red and blue lines and geographic framing should be given. If there are any significant points such as villages or mountain peaks, I would suggest highlighting them in this introductory figure and use them as reference points in subsequent plots.
    - Figure 2: unclear. I think it would be necessary to add reference points and represent the same observation area to have a better qualitative comparison of the two methods. If possible, I would implement the quality of the legend of the velocity values. Also, it is not clear what the blue line in Figure 2b means; a small legend would be needed. Are the red and green lines in Figure 2a the same as the blue and red lines in Figure 1? If so, maybe better to keep the same colors. What do the dots in Figure 2a represent?
    - Figure 5: for clarity of representation, I would suggest always keeping the north arrow upward (same as Figures 1 and 2); if possible, add reference points (e.g., location of villages) and the graphical scale of representation (even just in the first plot in the upper left corner). The legend has no units and does not indicate what kind of values are plotted.

We will make suggested edits to the figures mentioned above.

---

## Author Response (AR1)

**RC 1 Feedback**

1. **Feedback**: I recommend including a more detailed geomorphological, geological and stratigraphical description of the area interested by the El Forn landslide. For a complete characterization of a landslide through InSAR data, it is essential to investigate the geologic context in more detail to understand the conditions under which it is developing.
    a. **Response**: *Additional details about the geological makeup of the shearing surface of the landslide can be found in Section 2.1, with a more explicit direction to additional works containing a more in-depth geological description and analysis in previously-published works.*

2. **Feedback**: I recommend extending this InSAR analysis to the entire landslide body and not just the Cal Ponet-Cal Borronet lobe sector. Because chapter 2.1 describes the presence of 12 scattered boreholes in the landslide body for monitoring that should be exploited as a real opportunity for comparison with the InSAR data. The greatest strength of satellite interferometry is the ability to monitor large areas, here authors have focused only on a very small sector of a very large landslide, missing the most important information provided by the InSAR data.
    a. **Response**: *The other boreholes in question are monitored via analog non-continuous measurements (irregularly, approximately once per month), which is why we chose to work exclusively with S10. It is important to note that, while there is available analog data over the landslide, further investigation into available analog data provided only one data point as a viable option for a time-series comparison with InSAR. In order to not reduce the fidelity of the continuous time series from S10, the authorship chooses to not include this data in the body of this text. The authorship has made this data available upon request.*

3. **Feedback**: I recommend expanding the monitoring period of InSAR data, as the abstract specifies that Sentinel-1 data processed for 2019-2021 has been exploited, while Chapter 2.2 explains that interferograms from only a narrow time period between June and November 2019 were used. Again, this choice comes at the expense of one of the major strengths of the InSAR data, namely the possibility of providing long time series. Instead, focusing the analysis only to a 6-month time period and on a narrow area of the landslide appears as a serious limitation in the study. When analysing the behaviour of a landslide, it is a good practice to expand the analysis of the time series as much as possible in order to know as much information as possible.
    a. **Response**: *We have elected to include the yearly comparison of in situ and InSAR displacements with S10 as an attachment directly available in response for review. Since the landslide was arrested in 2019 (which we have stated more explicitly in the body of the work), we provide the raw data comparison of InSAR and in situ data and can include in the final version of the text upon request from review. Please see below.*

[Figure]

4. **Feedback**: I strongly recommend adding a chapter discussing the results before the conclusions. A chapter of discussion is essential for the explanation of the results and to understand applicability, advantages, and limitations of the proposed approach.

    a. **Response**: *We have included a more in-depth discussion, separate from the conclusion, in the revised body of the text.*

FIGURES:

- **Feedback, Figure 1:** a geographical localization is missing. North arrow between the large-scale image of the landslide and the focus are not in the same direction. I recommend pointing the north arrow upward. In addition, for the purpose of characterizing the study area please add the location of all 12 boreholes.

    o **Response**: *The orientation of the satellite images has been changed to fit the visuals of the results (due West), and Figure 1 and 2 have been changed to reflect this. Additionally, we have chosen to not include the location of every borehole as explicitly in this summary image but note that the general locations of the boreholes can be seen in Figure 2(a), generalized as red, green, and yellow dots (colors delineated by monitoring agency). As mentioned earlier in the text, we have chosen to refrain from adding comparison of other borehole readings due to their analog nature and as such we have chosen to not include in the introductory figure.*

- **Feedback, Figure 2:** the deformation map on the left has neither north arrow nor scale, also in the legend there is no explanation of what the coloured circles are. Moreover, the figure on the right is just a screen captured by EGMS: I recommend downloading the data and reshaping the figure (you can use the EGMS-stream application to download the data).

    o **Response**: *Figure 2(a) has been amended to reflect an arrow and scale. The dots in Figure 2a represent the individual data points available via the EGMS platform, and this has been made more explicit in the body of the text. At time of writing, the data has been masked out over the region of Andorra, in addition to the archive available through the EGMS platform.*

> *The authorship acknowledges these challenges, and the data may be made available upon request from the authorship.*

- **Feedback, Figure 4:** it is not clear why the representation of the landslide is now rotated 90°. I recommend defining a direction for the landslide representation and using it for all the figures in the paper.
    - **Response**: *The orientation of the satellite images has been changed to fit the visuals of the results (due West).*
- **Feedback, Figure 5**: the colour scale of the ordinary kriging results of various random samples are always different, this makes it so that an immediate visual comparison between the figures is not possible.
    - **Response**: *The legend bars have been removed for clarity purposes and the caption further elaborates what is being measured.  The main legend bar is kept in the original image for comparison. Additionally, the caption has additional context of what is being measured.*

**RC 2 Feedback**

1. **Feedback**: Although it is not the main focus of the paper, I believe that additional morphological and geological data of the study area should be provided. If available, the Authors should state in more detail the geological framework and the material involved in the landslide process, the geometry of the body, and the type of activity (e.g., periods of greatest acceleration). Additionally, if possible, it would be interesting to see the temporal evolution of some of the monitored data (e.g., displacements or piezometric level) and perhaps relate them to external factors such as rainfall. I think that providing a more extensive characterization of the landslide would allow an easier understanding of the data presented in the subsequent paragraphs, as well as the choices made; also, it will improve the quality of the paper. If the Authors do not consider it necessary, they may still better mention previous works carried out on the landslide body exploring these issues.
    a. **Response**: *Additional details about the geological makeup of the shearing surface of the landslide can be found in Section 2.1, with a more explicit direction to additional works containing a more in-depth geological description and analysis in previously published works.*
2. **Feedback**: In Section 3.2, I would suggest a better discussion of the choice of using a 6-month time interval to compare in situ observations and InSAR data. What are the reasons for this choice that do not allow for the analysis of annual trends? It could be the presence of snow cover or lack of monitoring data, either way, it needs to be specified and discussed in the text.
    a. **Response**: *Additional explanations for the 6-month interval have been made more explicitly in the body of the text.*
3. **Feedback**: Reading the paper, I found the lack of a general discussion highlighting the significance of the work, especially in terms of the use and applicability of the method. I think a "discussion section" could help to better emphasize the validity of the results obtained as well as the achievement of the goals. In my opinion, some scientific issues and questions that could be addressed in this section would be: What are the main advantages and limitations over other landslide monitoring methods? Can this method be applied in remote areas where there are no active monitoring systems? How can the method be improved and applied to implement an in situ monitoring network?
    a. **Response**: *We have included a more in-depth discussion, separate from the conclusion, in the revised body of the text. We have addressed the scientific issues and questions outlined by the reviewer in more detail within the discussion.*

FIGURES:

- **Feedback, Figure 1:** The orientation of the satellite images is not very clear; it would be better to point the north arrow upward to maintain the same visual angle in all the sketches presented. I find placing the EGMS point in this figure to

be not very intuitive; I would move it to other subsequent figures and insert here the location of the 12 boreholes mentioned in the text. Also, a small legend with the meaning of the red and blue lines and geographic framing should be given. If there are any significant points such as villages or mountain peaks, I would suggest highlighting them in this introductory figure and use them as reference points in subsequent plots.

- o **Response**: *The orientation of the satellite images has been changed to fit the visuals of the results. Additionally, a legend for the landslide and the smaller lobe (green and red) and labeling of the town have been added to provide more context for the reader. As mentioned earlier in the text, we have chosen to refrain from adding comparison of other borehole readings due to their analog nature and as such we have chosen to not include in the introductory figure. However, general locations of the boreholes can be seen in Figure 2(a), generalized as red, green, and yellow dots (colors delineated by monitoring agency).*

- **Feedback, Figure 2**: unclear. I think it would be necessary to add reference points and represent the same observation area to have a better qualitative comparison of the two methods. If possible, I would implement the quality of the legend of the velocity values. Also, it is not clear what the blue line in Figure 2b means; a small legend would be needed. Are the red and green lines in Figure 2a the same as the blue and red lines in Figure 1? If so, maybe better to keep the same colors. What do the dots in Figure 2a represent?

  - o **Response**: *The frames of reference for Figure 2 have been changed to match Figure 1, along with the outlines of the lobe and main scarp. The blue outline in Figure 2b represents the location of the El Forn landslide and has been made more clear in the caption. The dots in Figure 2a represent the individual data points available via the EGMS platform, and this has been made more explicit in the body of the text.*

- **Feedback, Figure 5**: for clarity of representation, I would suggest always keeping the north arrow upward (same as Figures 1 and 2); if possible, add reference points (e.g., location of villages) and the graphical scale of representation (even just in the first plot in the upper left corner). The legend has no units and does not indicate what kind of values are plotted.

  - o **Response**: *We appreciate the reviewer's suggestion for uniformity in the figure orientations for the landslide and have made the necessary changes so that Figures 1 and 2 have the same scale and orientation as the processed results (looking due West). Figure 5 has been adjusted to reflect orientation of the landslide, major landmarks, and velocities. The legend bars have been removed for clarity purposes and the caption further elaborates what is being measured. The main legend bar is kept in the original image for comparison.*

---

## Author Response (AR3)

**Author Response**

1. **Feedback**: The first one regards the effectiveness of the comparison they made between InSAR and borehole instrumentation. You still relied only on 1 borehole (continuously monitored borehole on the landslide), despite the presence of other 11 boreholes (manually reading). In my opinion, since the focus of you paper is the integration/comparison of in situ data and InSAR information, this activity must be expanded, event considering the lack of continuity in data acquisition. I consider this point important, because a proper integration could reveal deformation pattern and kinematics of the landslide.

    a. **Response**: *The authorship recognizes that additional points would be helpful to draw connection between in situ readings and InSAR. However, the in situ time frame for other borehole points do not offer readings that align within the time frame for the InSAR readings. More specifically, the analog readings only offer 1 time frame during the no-snow period of 2019. Please see the figures below indicating (1) the available data for several boreholes during this specific time period and the (2) location of these boreholes on the El Forn scarp.*

[Figure]

[Figure]

2. **Feedback**: The second one regards the quality of figures, that is still quite low. For instance: in Figure 1 localization of the landslide is still missing. I know the location of El Forn landslide, but it is fundamental to have a proper localization to support any potential reader interested to analyse the landslide. Amendment requested for figure 2 has been missed with an unclear reply on data availability. EGMS data (2015-2021) are available for download from the EGMS viewer.

   a. **Response**: *Figure 1: Additional localization has been provided by providing further context of location of the landslide in the figure itself, in addition to more information about the landslide itself in the main text. Additional information about the Cal-Ponnet-Cal Barronet lobe can be found in the following:*

      i. *Zhao, C. and Lu, Z.: remote sensing Remote Sensing of Landslides-A Review, https://doi.org/10.3390/rs10020279, 2018.*
      ii. *EuroConsult: Forn de Canillo | Euroconsult S.A., https://euroconsult.ad/en/highlights/forn-canillo.*
      iii. *Seguí, C. and Veveakis, M.: Continuous assessment of landslides by measuring their basal temperature, Landslides, 18, 3953–3961, https://doi.org/10.1007/S10346-021-01762-X/FIGURES/3, 2021.*

*These references have also been added in the main text.*

*The original amendment requested for Figure 2 has not been made per the request of the reviewer due to insufficient data over the area of Canillo. Since the original screen grab of the landslide was taken, the data over the entirety of Andorra has been removed from the EGMS landslide, including the 2015-2021 archive. Please see screenshot below for proof of lack of record.*

[Figure]